# Drug detoxification dynamics explain the postantibiotic effect

Jaydeep K Srimani[1], Shuqiang Huang[2], Allison J Lopatkin[1] & Lingchong You[1,3,4,*] (iD)

## Abstract

**The postantibiotic effect (PAE) refers to the temporary suppression of bacterial growth following transient antibiotic treatment. This effect has been observed for decades for a wide variety of antibiotics and microbial species. However, despite empirical observations, a mechanistic understanding of this phenomenon is lacking. Using a combination of modeling and quantitative experiments, we show that the PAE can be explained by the temporal dynamics of drug detoxification in individual cells after an antibiotic is removed from the extracellular environment. These dynamics are dictated by both the export of the antibiotic and the intracellular titration of the antibiotic by its target. This mechanism is generally applicable for antibiotics with different modes of action. We further show that efflux inhibition is effective against certain antibiotic motifs, which may help explain mixed cotreatment success.**

**Keywords** antibiotic tolerance; postantibiotic effect; systems biology
**Subject Categories** Microbiology, Virology & Host Pathogen Interaction; Quantitative Biology & Dynamical Systems
**Mol Syst Biol. (2017) 13: 948**

## Introduction

The postantibiotic effect (PAE) refers to the temporary suppression of bacterial growth following transient exposure to antibiotics. This transient inhibition has been observed since the first studies of penicillin against *Pneumococcus* and *Streptococcus* in the 1940s. Even after the antibiotic had been degraded by a penicillinase, the target populations exhibited a significant lag before resuming growth (Bigger, 1944; Parker & Marsh, 1946; Eagle, 1949; Eagle & Fleischman, 1950). Subsequent studies have observed PAE following treatment with a variety of antibiotics, including aminoglycosides (Zhanel & Craig, 1994), β-lactams (Hanberger *et al*, 1990; Odenholt-Tornqvist & Löwdin, 1991), fluoroquinolones (Athamna, 2004; Mizunaga, 2005), and others (Zhanel & Hoban, 1991; Odenholt-Tornqvist, 1993), and against both Gram-positive and Gram-negative bacterial species (Eagle & Musselman, 1949; Eagle *et al*, 1950; Bundtzen *et al*, 1981). PAE has also been observed in animal models (Craig, 1993; Gudmundsson & Einarsson, 1993), where, in addition to suppressing growth, transient antibiotic treatment can render the surviving population more susceptible to innate immune responses and result in decreased virulence expression (Eagle, 1949). Moreover, the extent of PAE is of vital importance in the design and optimization of periodic and multi-dose antibiotic regimens (Eagle *et al*, 1950; AliAbadi & Lees, 2000). For example, antibiotics that induce a long PAE can be dosed less frequently. However, the high concentrations required to reduce dosing frequency may result in adverse consequences, for example, toxicity (Avent *et al*, 2011). Accordingly, PAE is a standard metric used to evaluate novel antibiotics (Beam *et al*, 1992).

Despite widespread observations of PAE, its underlying mechanisms are not well established. Previous studies have speculated on a number of possible explanations, including nonspecific binding and nonlethal damage induced by antibiotic treatment (Craig & Vogelman, 1987; Li *et al*, 1997), antibiotic persistence within the periplasmic space, or the resynthesis of essential enzymes (MacKenzie & Gould, 1993). Moreover, these studies have not ruled out the possibility of multiple concurrent mechanisms, or different mechanisms being applicable for different antibiotics. More recently, zur Wiesch and colleagues proposed that antibiotic–target binding kinetics are sufficient to explain PAE (zur Wiesch *et al*, 2015), and fit their model to bacterial responses to tetracycline treatment.

Given that many, if not all, antibiotics lead to PAE, we asked whether there exists a common core mechanism that dictates the generation of PAE. In this study, we propose that a minimal unifying titration-based interaction, emphasizing target titration and antibiotic efflux, is sufficient to account for PAE observed in response to a wide variety of antibiotics. Our results indicate that efflux inhibition, an established antibiotic adjuvant strategy, may be effective only in conjunction with certain antibiotics. Moreover, understanding transient dynamics in antibiotic response is essential to designing effective combinations of drug and efflux inhibitor.

1 Department of Biomedical Engineering, Duke University, Durham, NC, USA
2 Center for Synthetic Biology Engineering Research, Shenzhen Institutes of Advanced Technology, Chinese Academy of Sciences, Shenzhen, China
3 Center for Genomic and Computational Biology, Duke University, Durham, NC, USA
4 Department of Molecular Genetics and Microbiology, Duke University School of Medicine, Durham, NC, USA
*Corresponding author. Tel: +1 919 660 8408; E-mail: you@duke.edu

# Results

To quantify population recovery dynamics, we first defined population recovery time ($RT_{pop}$) as the doubling time relative to the end of antibiotic treatment (Fig 1A, left panel). PAE, then, corresponds to the prolonged recovery time in response to antibiotic treatment, as compared to the control (Fig 1A, right panel). In the absence of PAE, populations would resume normal growth immediately after the antibiotic was removed. The corresponding recovery time would be independent of the antibiotic treatment. This metric is preferable to typical definitions of PAE, which measure the time required for a population to increase 10-fold after treatment via counting colony-forming units (CFUs) (Eagle & Musselman, 1949);

by measuring twofold increases in cell density, our metric more precisely captures the effect of transient population dynamics immediately following antibiotic treatment.

In addition to potentially masking recovery dynamics on short timescales, previous studies of PAE [including those quantifying recovery by measuring rates of ATP synthesis (Hanberger *et al*, 1990) and DNA and protein synthesis (Stubbings, 2006)] suffer from a common disadvantage: Due to sparse time series data, they do not yield temporally precise estimates of recovery time. Moreover, they rely on relatively large populations of cells, which could potentially lead to inoculum effects and skew recovery dynamics (Tan *et al*, 2012). To overcome these technical limitations, we used a custom-made microfluidic device to quantify the response of bacterial

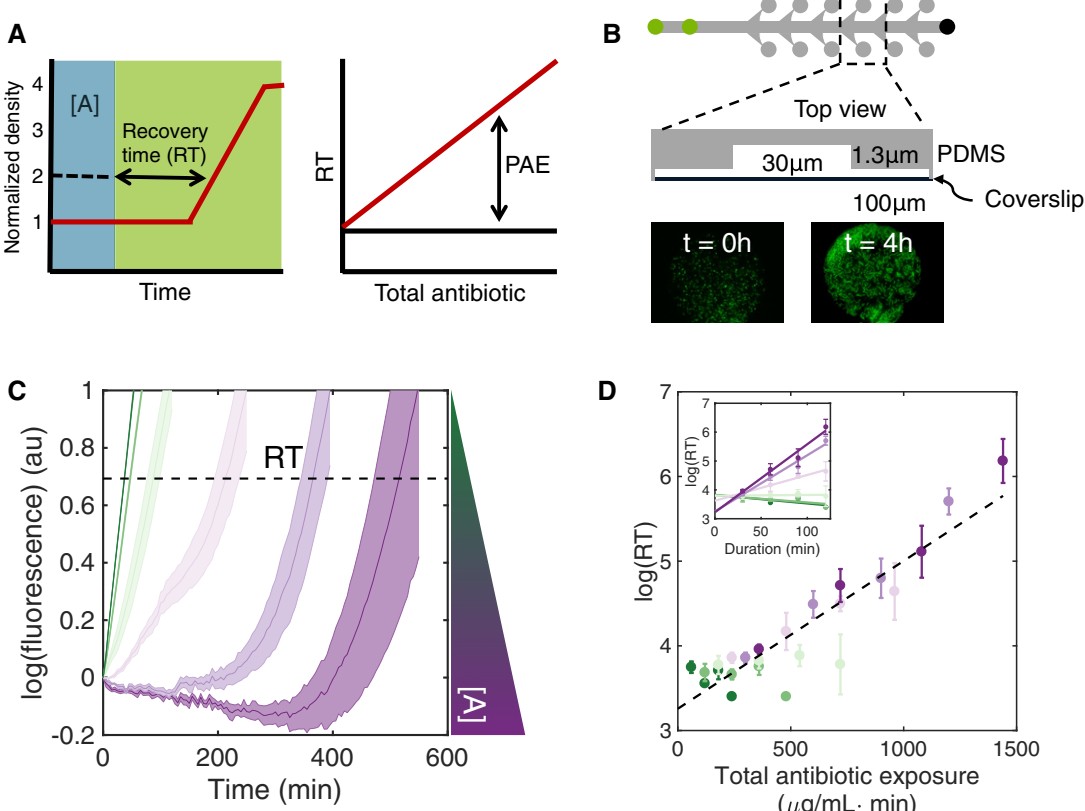

**Figure 1.  Recovery time increases exponentially as a function of total antibiotic exposure.**

A  (Left panel) We define the recovery time (RT) as the time required for a population (red line) to double in response to a transient antibiotic treatment (blue shading). (Right panel) The postantibiotic effect (PAE) induced by an antibiotic treatment refers to the additional time required for a population to recover (red line) in comparison with the untreated control (black line).

B  A microfluidic device for quantitative recovery time measurements. Each PDMS-fabricated chip consists of six independent channels, one of which is shown, from top view (top row). Green circles indicate media inflow; black circle indicates outflow. Bacteria are manually loaded in inflow ports and are trapped in individual culturing chambers (gray circles), and the height of which ensures that bacteria are imaged in a monolayer (bottom row). Growth conditions (e.g., antibiotic dose profile) are controlled via programmable syringe pumps. Fluorescent images (bottom row) show representative growth of a monolayer of *Escherichia coli* BW25113 cells constitutively expressing GFP over four hours.

C  Representative time series fluorescence data showing dose-dependent population recovery in response to transient streptomycin treatment. Here, time zero corresponds to the end of treatment (120 min). Trajectories show mean and standard deviation for five replicates; colors indicate increasing antibiotic concentration (2, 4, 6, 8, 10, 12 µg/ml). Fluorescence values are normalized to those at time zero. Dotted line indicates a twofold increase, corresponding to the recovery time for each population.

D  Recovery time increases with total antibiotic exposure. Inset shows recovery time as a function of dose duration for increasing streptomycin concentrations (as in panel C). When plotted against total antibiotic exposure (calculated as $\int_o^D A(t)dt$), all recovery time values collapse onto a single exponential function (black line, $R^2 = 0.83$). Recovery time was measured in response to streptomycin concentrations of 0, 2, 4, 6, 8, 10, and 12 µg/ml, and treatment durations 30, 60, 90, and 120 min. Error bars indicate standard deviation, calculated from five replicates; *y*-axis uses the natural logarithm.

populations to a range of antibiotic concentrations and treatment durations (Lopatkin *et al*, 2016). As shown in Fig 1B, the microfluidic chip consists of six independent channels (top row, top view), each with two media inputs (green circles) and one output (black circle). Bacteria are manually loaded into each channel and trapped in individual cylindrical culturing chambers (middle row). The height of the chambers (~1.0 µm) constrains bacterial growth to a single monolayer, facilitating precise quantification (bottom row) using time-lapse fluorescence microscopy. Growth media and antibiotic dosing protocols are controlled by programmable syringe pumps.

Figure 1C shows typical time courses of *Escherichia coli* strain BW25113 (Grenier *et al*, 2014) constitutively expressing GFP, exposed to 120-min streptomycin treatment at increasing concentrations (where $t = 0$ corresponds to the end of treatment, and the observed $IC_{50}$ for streptomycin was 1.99 µg/ml). We found that the fluorescence signal serves as a reliable surrogate measure of the population density over the range of dosing conditions used (Appendix Fig S1). It is particularly suited for the imaging-based characterization of the antibiotic response in the microfluidic device. The recovery time drastically increased with increasing antibiotic concentrations, demonstrating the generation of PAE, where the dashed line indicates a twofold increase relative to $t = 0$. We note that for high streptomycin concentrations, the population fluorescence decreases before recovery; this is likely due to continued inhibition after the removal of antibiotic. At a fixed concentration, the recovery time drastically increased with treatment duration (Fig 1D, inset). Remarkably, by combining the dose and duration for each treatment, we found that the recovery time was determined by the total antibiotic exposure for each treatment, regardless of the dose profile (Fig 1D, main). That is, every treatment (i.e., combination of concentration and duration) that delivered a given total antibiotic resulted in a comparable recovery time, which increased approximately exponentially with the total antibiotic exposure.

A potential caveat of using fluorescence reporters is the effect of incomplete population death during antibiotic treatment. Indeed, under some treatment conditions, we observed cells that expressed GFP at comparable levels to other population members but did not divide for the duration of the recovery phase (~18 h). Therefore, although these cells contributed to total fluorescence, they did not contribute to population recovery. To account for this, we measured population viability as a function of streptomycin concentration and duration in the absence of growth (Appendix Fig S2A). We adjusted the twofold cutoff to account for this death and ensure that doubling times reflect the signal from only viable cells; PAE remained a function of total antibiotic exposure (Appendix Fig S2B).

To gain insight into generation of PAE, we adopted a kinetic model of antibiotic-mediated inhibition of ribosomes (e.g., streptomycin; Tan *et al*, 2012) (Fig 2A). The model consists of six ordinary differential equations (ODEs) that account for the transport of the antibiotic across the cell membrane ($A_{in}$ or $A_{out}$), synthesis of the ribosome ($C$) through a positive feedback loop, and antibiotic-mediated ribosome inhibition and potential degradation (equations 4–9 and Appendix Table S1). The ribosome concentration $C$ sets the population growth rate $\mu$ (Neidhardt, 1996), connecting the intracellular drug–cell interaction with overall population recovery. Here, we assume homogenous populations that interact (i.e., drug influx and efflux) with a common extracellular environment.

Because the population growth rate is dependent on the ribosome concentration, we can investigate recovery on two scales: on the population level ($RT_{pop}$, analogous to Fig 1D) as the time required for the cell number to double, and on the individual level, as the time required for the ribosome concentration to achieve its half-maximal synthesis rate ($RT_{cell}$). In terms of the core structure, our model is similar to one developed by zur Wiesch *et al* (2015), but provides a more detailed description of the underlying kinetics.

Figure 2B shows the typical dynamics of various components, in three phases, in response to the addition and removal of an extracellular antibiotic. During antibiotic treatment (Fig 2B, left schematic and blue shading), $A_{out} \gg A_{in}$ and the dominant dynamic is the accumulation of intracellular antibiotic due to influx. Neglecting the efflux dynamics and titration due to binding of antibiotic to the ribosome, and denoting the dose duration as $D$, we have: $A_{in}(D) = \frac{k_{in}}{k_{out}} A_{out}(1 - e^{-k_{out}D}) \approx k_{in} A_{out} D$ (for sufficiently small $D$). That is, at the end of an antibiotic pulse, *the intracellular antibiotic concentration is proportional to total amount of the antibiotic used* (Fig 2C, panel i), consistent with previous studies (Hancock, 1962; Hurwitz & Rosano, 1962; Bryan & Van Den Elzen, 1976; Damper & Epstein, 1981; Muir *et al*, 1984).

Upon removal of the antibiotic, $A_{in} \gg A_{out} = 0$ and the transport dynamics of the antibiotic are dominated by efflux (Fig 2B, middle schematic and green shading). In addition to efflux, binding kinetics between ribosome and antibiotic serve as an intracellular reservoir for $A_{in}$ and can further delay recovery. If the cell survives treatment, it can revert to a basal growth rate when it is sufficiently detoxified (i.e., when $A_{in}$ is sufficiently small to not inhibit the ribosome effectively) (Fig 2C, panel ii). The balance between these three processes (efflux, binding, and degradation) sets the timescale of detoxification and subsequent growth, leading to generation of PAE. Furthermore, because $C \ll A_{in}$, efflux dominates this interplay. Consistent with this notion, for different dosing protocols, the correlations between $C$ and $A_{in}$ approximately collapse to a single line (Fig 2D). This mechanism recaptures the dependence between population recovery time and the total antibiotic exposure (Fig 2E). Moreover, modeling indicates that this dependence is independent of initial cell density (Appendix Fig S3). We note that this model assumes that antibiotic is freely transported across the cell membrane and binds to targets either in the cytoplasm or periplasm, as opposed to those that damage the cell membrane. Although inoculum effects could potentially be induced by these antibiotics, the range of treatment conditions used ensure that all populations undergo first inhibition followed by recovery.

These results provide a simple explanation for the emergence of PAE: Fundamentally, PAE arises as the time required for individual cells to recover by exporting antibiotic, such that ribosome-mediated positive feedback can be activated. This "detoxification" process occurs on the individual cell level, and intracellular recovery dynamics correlate strongly with population recovery (Fig 2F, $R^2 = 0.86$). Moreover, this correlation is robust to changes in the antibiotic-mediated ribosome degradation rate (Appendix Fig S4). Here, we do not consider antibiotic dilution by cell growth, which would result in lower recovery times without changing our qualitative conclusions.

A mechanistic understanding of PAE is critical, as recovery time to a single dose of antibiotic represents a critical metric that can predict the efficacy of long-term multi-dose treatments using the

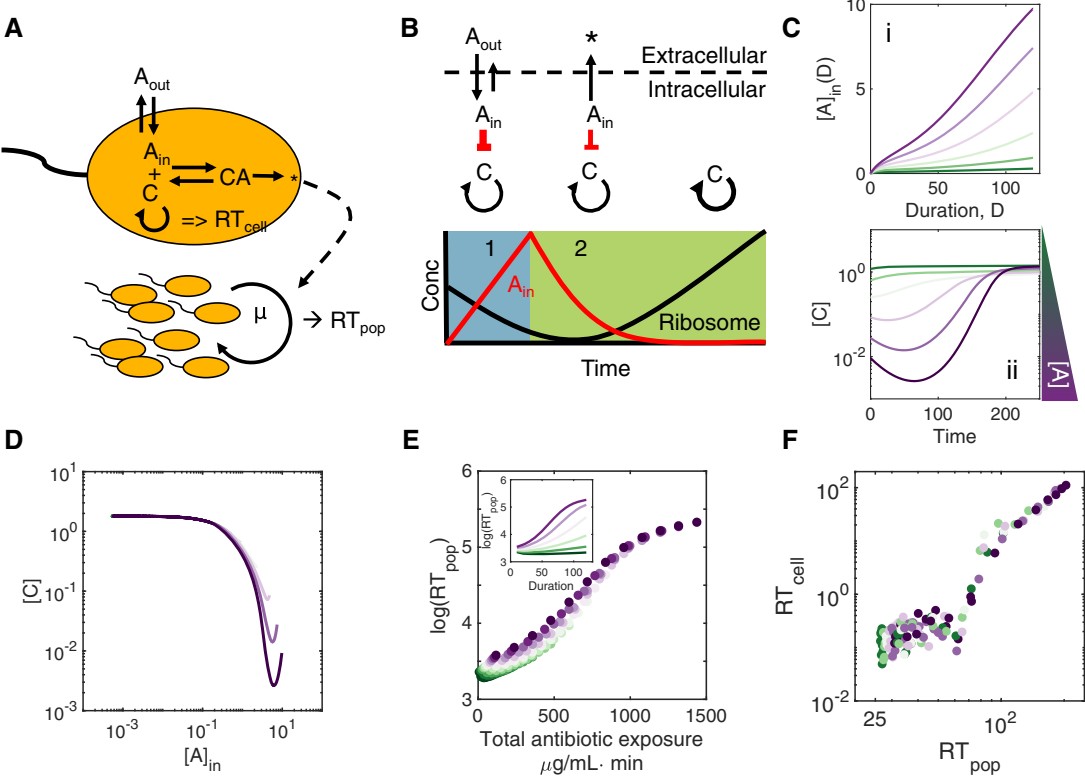

**Figure 2. The timescale of overall intracellular antibiotic dictates individual and population recovery.**

A   A minimal model of antibiotic action. Antibiotic is transported between the intracellular and extracellular environments ($A_{in}$ and $A_{out}$) by influx and efflux rates ($k_{in}$ and $k_{out}$). $A_{in}$ reversibly binds to target ribosomes $C$ to form the complex $CA$, with binding and dissociation rates $k_f$ and $k_b$, respectively. This complex can then be degraded through the intermediate $CA'$. These intracellular dynamics influence the overall population recovery rate; the maximum growth rate $\mu$ is dependent on the ribosome concentration $C$. Thus, recovery can be quantified both on the individual level, as a function of $C$, or on the population level, in terms of cell density.

B   Concurrent antibiotic transport and ribosome inhibition dictate recovery dynamics. During treatment, $A_{out} \gg A_{in}$, and the intracellular antibiotic concentration can be linearly approximated by $A_{in}(D) \approx k_{in}A_{out}D$, where $A_{out}$ is assumed to be constant and $D$ represents the treatment duration (leftmost schematic and panel 1). Antibiotic influx is greater than efflux; $A_{in}$ binds to target ribosomes and strongly inhibits the upregulation of ribosome synthesis. When extracellular antibiotic is removed, $A_{in}$ decreases (middle and rightmost schematics, green shading and panel 2), and ribosome synthesis resumes when $A_{in}$ is sufficiently small.

C   $A_{in}$ accumulates linearly with treatment duration for sufficiently short treatment durations (panel i). After the removal of $A_{out}$, efflux and inhibition dynamics combine to delay the synthesis of ribosomes in a concentration-dependent manner (panel ii). Colors indicate increasing antibiotic concentration, as shown in panel ii.

D   Antibiotic turnover timescale sets intracellular recovery $RT_{cell}$. Regardless of the antibiotic treatment history, the relationship between intracellular antibiotic $A_{in}$ and ribosome concentration $C$ approaches the same asymptote, indicating that the timescale of individual detoxification sets the timescale of ribosome synthesis. Colors indicate increasing antibiotic concentration, as in Fig 2C; for these representative trajectories, treatment duration was set to 120 min.

E   Population recovery time is dictated by total antibiotic exposure. Inset shows that $RT_{pop}$ is an increasing function of dose duration; these data collapse onto a single relationship as a function of total antibiotic exposure (main figure).

F   Individual ($RT_{cell}$) and population recovery time ($RT_{pop}$) are strongly correlated, suggesting that intracellular dynamics lead to population level recovery ($R^2 = 0.86$).

same antibiotic (Meredith *et al*, 2015). Here, periodic treatments are specified by concentration $A$, the dose duration $T_1$, the interval between doses $T_2$, and the total number of doses $N_{dose}$ (Appendix Fig S5A). $RT_n$ is defined as the recovery time in response to a given $N_{dose}$ treatment; larger values of $RT_n$ correspond to treatment success (Appendix Fig S5B). Appendix Fig S5C shows a transition in treatment outcome at $\frac{T_2}{RT_1} \approx 1$. Specifically, any treatment that delivers successive doses more frequently than the corresponding single dose recovery time ($RT_1$) will induce a significant $RT_n$. However, regardless of the frequency of antibiotic dosing, population recovery time remains a function of total antibiotic exposure (Appendix Fig S5D).

The model also predicts the consequences of modulating key processes (Fig 3). In particular, decreasing the rate of antibiotic-mediated ribosome degradation, increasing the efflux

rate, or increasing ribosome synthesis rate all lead to more rapid recovery for the same total antibiotic exposure; microfluidic experiments confirmed these model predictions. To investigate the effect of ribosome degradation, we compared the recovery time induced by streptomycin, which has been shown to induce the heat-shock response (HSR) and lead to rapid ribosome degradation, to chloramphenicol ($IC_{50}$  1.2 μg/ml), which does not induce HSR (Tan *et al*, 2012), and therefore induces slow ribosome degradation. Consistent with modeling predictions, chloramphenicol treatment resulted in significantly shorter recovery time than streptomycin treatment (Fig 3A and D shows modeling and experimental results, respectively).

To inhibit efflux pump activity, we used carbonyl cyanide 3-chlorophenylhydrazone (CCCP), a phosphorylation inhibitor that inhibits a variety of efflux pumps that rely on the proton motive

force (PMF), including resistance nodulation division pumps and others (Kinoshita *et al*, 1984; Cohen *et al*, 1988; Singh *et al*, 2011). We confirmed the concentration-dependent efflux inhibition (Appendix Fig S6). Modeling predicted that inhibiting efflux pump activity would result in longer recovery times (Fig 3B); therefore, we tested the effect of a subinhibitory CCCP concentration with chloramphenicol treatment, which alone induced a minimal recovery delay (Fig 3D). The addition of CCCP indeed caused significantly longer recovery times (Fig 3E); however, CCCP alone did not increase population recovery time.

Finally, we modulated the ribosome synthesis rate by modulating the richness of the growth media, specifically by adjusting the concentration of casamino acids: A higher concentration of the casamino acids leads to faster bacterial growth, all else being equal (Lazzarini & Dahlberg, 1971). Indeed, with faster growth, corresponding to rapid ribosome synthesis, populations recovered significantly faster after streptomycin treatment (Fig 3C and F shows modeling and experimental results, respectively).

These results suggest that drug response dynamics can be modulated to induce longer PAEs, which is advantageous from a clinical

perspective (Spivey, 1992; Fishman, 2006; Talpaert *et al*, 2011); antibiotic efflux inhibition is an active area of research (Marquez, 2005; Askoura *et al*, 2017). In particular, resistance nodulation division (RND) pump systems are prevalent in Gram-negative species and have been shown to efflux a variety of antibiotics, including penicillins and cephalosporins, macrolides, aminoglycosides, fluoroquinolones, and tetracyclines (Ma *et al*, 1994; Nikaido, 1994, 1998). Moreover, many resistant strains have been shown to overexpress these efflux pumps (Okusu *et al*, 1996; Mcmurry & Oethinger, 1998; Kriengkauykiat *et al*, 2005). In this context, efflux pump inhibitors (EPIs) have emerged as potential adjuvants for antibiotic treatment. Recent studies have proposed a number of candidate EPIs, via both rational design (Amaral *et al*, 2007) and natural isolation (Stavri *et al*, 2007); these are often based on studies of efflux pump structure, competitive binding, or disruption of transmembrane electrical gradients (Poole & Lomovskaya, 2006; Mahamoud *et al*, 2007). Intuitively, we would expect that efflux inhibition would be universally beneficial in lengthening PAEs in response to a given antibiotic treatment. However, these approaches have been met with limited clinical success (Van Bambeke & Lee, 2006; Opperman & Nguyen,

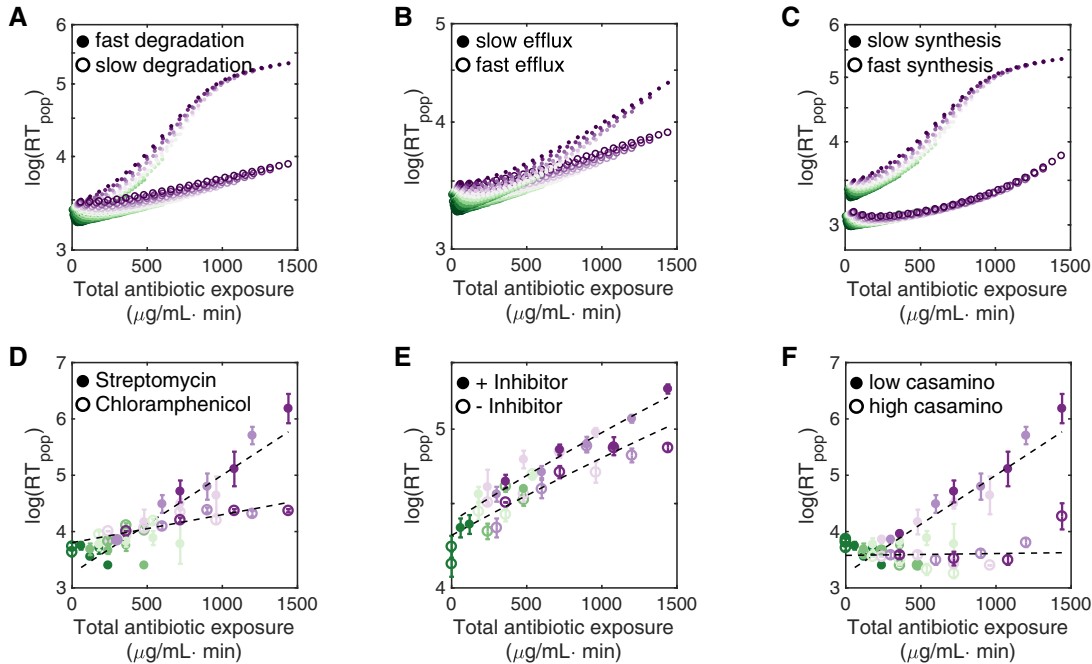

**Figure 3.  Key parameter perturbations confirm model validity.**

A  Modeling predicts that decreasing the ribosome degradation rate leads to shorter recovery times in response to equal amounts of total antibiotic exposure. Here, we use $k_d$ = 0.1 and $k_d$ = 0.2 for high and low degradation, respectively.

B  Modeling predicts that decreasing the antibiotic efflux rate leads to longer recovery times in response to equal amounts of total antibiotic exposure. Here, we use $k_{out}$ = 0.01 and $k_{out}$ = 0.001 for fast and slow efflux, respectively.

C  Modeling predicts that increasing the ribosome synthesis rate leads to shorter recovery times in response to equal amounts of total antibiotic exposure. Here, we use $k_1$ = 0.2 and $k_1$ = 0.4 for slow and fast ribosome synthesis, respectively.

D  Streptomycin treatment (closed data points, $R^2$ = 0.83) results in significantly longer recovery times than chloramphenicol treatment (open data points). Here, the difference between two responses is statistically significant ($P < 0.05$ by ANOVA). Error bars indicate standard deviation of five replicates.

E  The addition of efflux pump inhibitor (CCCP) (closed data points, $R^2$ = 0.91) increases population recovery time in response to chloramphenicol treatment (open data points, $R^2$ = 0.80). Here, CCCP was added at subinhibitory concentrations (3 μg/ml); in the absence of antibiotic treatment, CCCP alone did not inhibit population recovery. The CCCP-mediated increase in recovery time is statistically significant ($P < 0.01$ by ANOVA). Error bars indicate standard deviation of five replicates.

F  The ribosome synthesis rate was increased by increasing the concentration of the casamino acids in the media from 0.01% w/v (closed data points) to 0.05% w/v (open data points). Faster synthesis resulted in lower recovery times ($P < 0.001$ by ANOVA) in response to streptomycin treatment. Error bars indicate standard deviation of five replicates. In each of these panels, the color scheme indicates increasing antibiotic concentration, as in Fig 1C.

2015); optimal cotreatment strategies remain an open question (Lomovskaya & Bostian, 2006).

Therefore, we asked the question, under what conditions is efflux inhibition an effective strategy for extending PAE? We used modeling to examine the sensitivity of various parameters (e.g., ribosome degradation and synthesis rates, as well as positive feedback strength), and antibiotic motifs, to changes in efflux. We reasoned that the induction of PAE by antibiotic accumulation and detoxification dynamics (Fig 2A) is applicable to any drug that has an intracellular target (including those in the periplasm). As such, we expect that qualitatively similar dynamics can lead to PAE for these other antibiotics, and may be sensitive to efflux inhibition.

To test this hypothesis, we developed simplified kinetic models of intracellular dynamics to investigate three motifs of antibiotic action that encompass a wide variety of common antibiotics (equations 10–12 and Appendix Table S2). In each case, the antibiotic is transported between the intracellular and extracellular spaces. In the intracellular space, the antibiotic binds to its target reversibly and the concentration of the free target indicates the viability of the cell. However, these different motifs differ in how the target is regulated (corresponding to a change in one relevant parameter in each case). In the first motif, the target synthesis is driven by a positive feedback loop, and binding to the antibiotic leads to enhanced degradation of the target (Fig 4A, left column). This motif accounts for antibiotics that target ribosome synthesis and induce fast ribosome degradation, including aminoglycosides such as streptomycin and kanamycin. The second motif is identical to the first except that the antibiotic does not enhance degradation of the target (Fig 4A, middle column). This motif accounts for the action by chloramphenicol and tetracycline (Tan *et al*, 2012), as well as fluoroquinolones such as ciprofloxacin. The third motif is identical to the second except that the target is synthesized at a constant rate, with no feedback (Fig 4A, right column). This motif can account for inhibition by β-lactams. For each of these simplified motifs, recovery time remains a function of total antibiotic exposure (Fig 4A, second row). However, decreasing efflux rates (Fig 4, open circles) results in a significant increase in recovery time for the aminoglycoside motif only (Fig 4A, left panel). Efflux inhibition was less effective when target degradation was negligible (middle panel) or the target did not undergo positive feedback (right panel). These results suggest that PAE is dictated by total antibiotic exposure and minimal binding/transport rates, independent of an antibiotic's specific mechanism of action. Rather, target binding and efflux are critical processes underlying this relationship.

To quantify the effects of efflux inhibition, we define sensitivity as the total change in recovery time, over a range of antibiotic doses, in response to a change in efflux rate (Appendix Fig S7A). With this definition, we first examined the effect of efflux inhibition in conjunction with the drug-mediated ribosome degradation rate (Appendix Fig S7B). We observed that faster degradation resulted in increased efflux sensitivity. Similarly, we found that increasing the nonlinearity of the target positive feedback loop also increased efflux sensitivity (Appendix Fig S7C). These results suggest that the efficacy of efflux inhibition as an adjuvant treatment depends on the particular antibiotic used.

To test the model predictions, we used an *E. coli* strain constitutively expressing bioluminescence (Andreu *et al*, 2010) (see

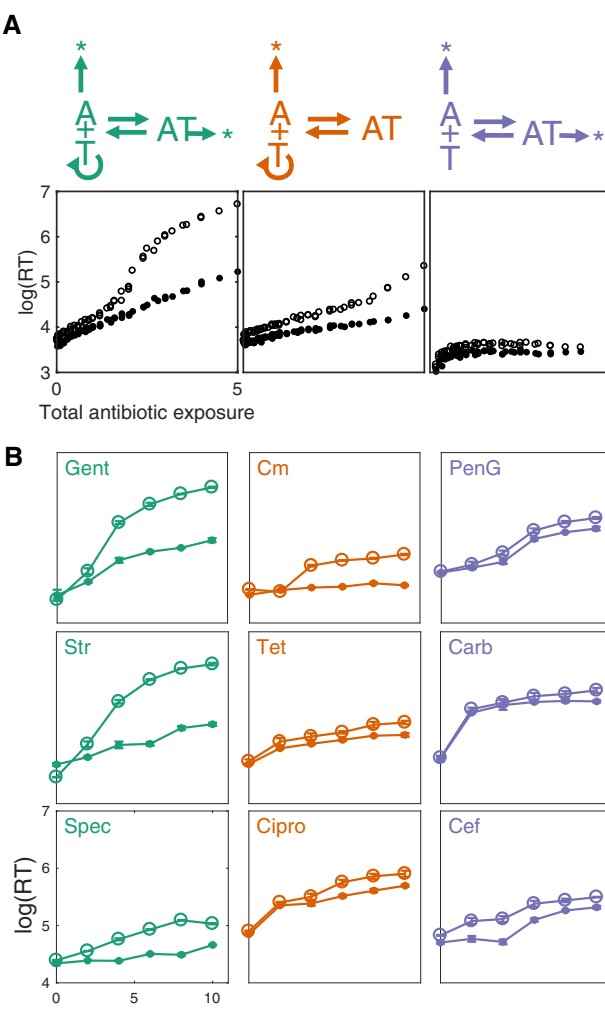

**Figure 4. Efflux inhibition is an effective cotreatment strategy for certain antibiotics.**

A  Three general motifs of intracellular antibiotic action. Left and middle motifs correspond to ribosome-inhibiting antibiotics that induce rapid and minimal ribosome degradation, respectively. In these motifs, the target molecule is subject to nonlinear positive feedback (i.e., transcription and translation, in the case of ribosomes). Right motif corresponds to antibiotics that inhibit other targets that are not subject to positive feedback, for example, β-lactams. In each case, recovery time is a function of total antibiotic exposure (closed circles) and inhibiting efflux results in longer recovery times (open circles).

B  Inhibiting antibiotic efflux with CCCP significantly increased recovery time for antibiotics that induced rapid target degradation and involved a positive feedback loop. Antibiotics are as follows: penicillin G (PenG), spectinomycin (Spec), gentamicin (Gent), streptomycin (Str), chloramphenicol (Cm), tetracycline (Tet), and carbenicillin (Carb). Antibiotic concentrations used have been scaled by their respective IC$_{50}$ values (solid points and open circles show response with no efflux inhibition and 2 μg/ml CCCP, respectively). Colors correspond to the motifs of action shown in Fig 4A.

Appendix Supplementary Methods for full details), which is commonly used as a reporter of viable cell density (Prosser *et al*, 1996), accurately reflects cell density over the full range of conditions tested (Appendix Fig S1B), and bypasses the need to control

for antibiotic-induced partial population death, as in Fig 1. This reporter allows us to test PAE in a higher throughput compared to the microfluidic device by using a microplate reader. We tested a number of antibiotics (See Appendix Table S3) over the same range of $IC_{50}$ values (Appendix Fig S8). Although the addition of CCCP at a subinhibitory concentration (2 µg/ml) increased the extent of PAE in all cases (Fig 4B), this effect was most significant with the three aminoglycoside antibiotics (streptomycin, gentamicin, and spectinomycin), as compared to drugs that fall in either of the latter two motifs. These results confirm our modeling predictions; the three aforementioned antibiotics induce rapid ribosome degradation. Moreover, we observed minimal inhibition with the β-lactams penicillin G, carbenicillin, and cefotaxime, as suggested by sensitivity analysis.

## Discussion

In general, there are two strategies to combat the rapid rise of drug-resistance microbial pathogens: novel antibiotic development or more effective use of existing drugs (Fishman, 2006; Kaki et al, 2011). Given the prohibitive time and financial costs of the former, the latter strategy is becoming increasingly critical. In doing so, it is important to move beyond steady-state measures of efficacy, for example, $IC_{50}$, and emphasize the importance of temporal dynamics in drug response. PAE is one such response; our results suggest that a minimal but common motif is sufficient to account for this phenomenon in response to a wide variety of antibiotics. These results suggest that efflux-mediated recovery could be a unifying motif to explain PAE in response to a wide variety of antibiotics. This would tie together wide-ranging literature reports of PAE in many antibiotic–bacterium combinations, without requiring distinct drug-dependent explanations. Here, we note that, in addition to antibiotic binding and transfer dynamics, dynamics of PAE in vivo are dependent on pharmacokinetic/pharmacodynamics (PK/PD) factors including availability, toxicity, and plasma vs total drug concentrations (Toutain et al, 2002). Indeed, in vitro PAE measurements often underestimate in vivo observations (Renneberg & Walder, 1989). Optimal antibiotic dosing treatments will need to take these considerations into account to ensure that PAE remains predictive of treatment outcome (Vogelman et al, 1988; Papich, 2014).

Our analysis represents a parsimonious explanation of PAE, based on the combination of the binding kinetics between an antibiotic and its target and the transport of the antibiotic. We do not consider nonspecific binding to secondary antibiotic targets, or drug dilution by cell growth (zur Wiesch et al, 2015). While these factors are not essential to the underlying relationship between antibiotic dosing parameters and recovery, they would affect the extent of PAE: Nonspecific binding would enhance PAE; drug dilution by growth would attenuate PAE.

While these effects can contribute to, and modulate the extent of, PAE, they are not essential to its emergence. Indeed, cell division is extremely slow during detoxification; therefore, in our model, drug dilution exerts minimal effects. On the other hand, nonspecific binding would increase recovery time in response to a given amount of antibiotic; we focus on the minimal network dynamics that explain PAE for the widest possible variety of antibiotics. Furthermore, our model assumes homogeneous populations

and no cell death during dosing. Finally, intra-population variability in cell division and death rates, both during and after antibiotic treatment, can influence PAE (Gottfredsson & Erlendsdóttir, 1998; Wiuff et al, 2005). In this case, antibiotic selection would enrich more tolerant subpopulations and result in lower recovery times. However, accounting for biochemical noise does not qualitatively change our conclusions.

Moreover, we show that efflux inhibition is a more effective strategy to induce PAE for certain antibiotic mechanisms of action, namely those that rapidly degrade a target subject to nonlinear positive feedback, than for others. These observations provide a potential guidance for the use of EPIs as antibiotic adjuvants. Indeed, efflux-mediating adjuvants could themselves be used in combination, provided that concentrations used remain subinhibitory. However, rather than using them in all cases, it is vital to understand the transient population dynamics induced by particular antibiotics to determine whether EPI usage would result in a positive outcome.

## Materials and Methods

### Strains, media, and growth conditions

Unless otherwise noted, E. coli strain BW25113 ($F^-$, DE(araD-araB) 567, lacZ4787(del)::rrnB-3, LAM⁻, rph-1, DE(rhaD-rhaB)568, hsdR514) was used throughout this study. Unless otherwise noted, all experiments were conducted in M9 media supplemented with 0.4% w/v glucose and 0.1% w/v casamino acids, and at 37°C unless otherwise indicated. BW25113 cells were transformed with a constitutively expressed GFP plasmid (kanamycin resistant) for microfluidic and viability experiments; all cultures were supplemented with 50 µg/ml kanamycin for selection purposes. For luminescence assays, BW25113 was transformed with a constitutively expressed luminescence reporter plasmid (kanamycin resistant; Andreu et al, 2010). For all experiments, overnight cultures were grown from single colonies, picked from streaked plates, in 3 ml Luria-Bertani (LB) broth for 16 h at 37°C with 250 rpm shaking. Streaked plates were stored in 4°C when not in use, and remade from glycerol stocks every 2 weeks.

### Fluorescence/luminescence reporter calibration

Overnight cultures were diluted 100-fold into fresh M9 media containing selecting antibiotics (kanamycin, 50 µg/ml). Cells were grown in 3 ml aliquots in the presence of increasing streptomycin concentrations (0, 2, 4, 6, 8, 10, 12 µg/ml) for 2 h at 37°C and 250 rpm shaking. Following treatment, cells were serially diluted and plated on selective agar plates to measure CFU counts; GFP fluorescence and luminescence were measured in 96-well black-walled plates (Corning) using a Tecan InfinitePro M200 plate reader (GFP emission/excitation 485/535 nm). Plates were incubated overnight at 37°C and CFUs counted the following morning. All measurements were done in quadruplicate.

### Microfluidic platform fabrication and experimental protocol

Transient antibiotic dosing experiments were carried out using a previously published microfluidic platform (Lopatkin et al, 2016).

Briefly, microfluidic chips were fabricated with polydimethylsiloxane (PDMS) using a silicon mold. Each chip is comprised of six parallel replicate units, each of which consists of a central flow channel and 24 branched culturing chambers. The height of these chambers (~1.3 μm) traps a monolayer of bacterial cells, thereby enabling accurate image processing and population quantification. Each flow channel has two inputs and one output; by controlling media conditions (flow rate, antibiotic concentration) using externally programmed syringe pumps (New Era Pump Systems NE-1600), various antibiotic dose profiles can be implemented.

Overnight cultures were diluted 100-fold into 3 ml M9 and grown for 2 h at 37°C with 250 rpm shaking, reaching an $OD_{600}$ of ~0.2. Cells were then condensed 10-fold. Chips were briefly vacuumed to create negative pressure, which enabled cells to be manually injected into an input using a P2 pipette. Each experiment started with roughly 100 cells per culturing chamber. Media flow in the chip was controlled with syringe pumps. Cells were allowed to grow for 30 min in clean media flow (no antibiotics, 120 μl/h flow rate), followed by antibiotic treatment (500 μl/h with applicable antibiotic concentrations) and a recovery phase (no antibiotics, 120 μl/h flow rate). These flow rates were selected to satisfy two physical limits: Excessively high flow rates resulted in cell washout, and insufficiently low flow rates resulted in minimal antibiotic effect over the range of dose durations used. The entire device was kept at 37°C. Five replicates were used per experimental condition. Culturing chambers were imaged every 5 min using a DeltaVision Elite deconvolution microscope.

### Image processing and analysis

All image data from microfluidic experiments were analyzed using custom MATLAB scripts. Following background subtraction, total fluorescence was quantified and normalized to the initial time point following antibiotic treatment. Figures show mean and standard deviation for five replicates.

### Quantifying population viability

Overnight cultures were regrown to mid-exponential phase by 100-fold dilution in 3 ml M9 media and incubating for 2 h at 37°C with 250 rpm shaking. Cells were then washed and resuspended in a standard 1× M9 buffer (M9 not containing glucose or casamino acids); 500 μl aliquots in 1.5-ml Eppendorf tubes were incubated with appropriate antibiotic concentrations at 37°C with 250 rpm shaking. Samples were then serially diluted to $10^7$-fold and plated on selecting agar plates (50 μg/ml kanamycin). Plates were incubated at 37°C overnight and CFUs counted the next day. Cell density was also measured prior to antibiotic treatment (~7E8 CFU/ml). All data points show the mean and standard deviation of six replicates. Viability curves for each streptomycin concentration were normalized to initial densities and fit exponential decay curves. The time constants of these fits were used to account for population death.

### Accounting for population death in microfluidic experiments

Viability curves were used to adjust the twofold cutoff used to calculate recovery time. As an example, suppose a particular antibiotic treatment killed a fraction ($X$) of cells. Then, the population

fluorescence immediately following antibiotic treatment can be expressed as a combination of dead and viable cells: $XF_0 + (1 - X)F_0$, assuming dead cells contribute equally to the fluorescence signal. Therefore, when the viable fraction doubles, the total fluorescence would be $XF_0 + 2(1 - X)F_0 = (2 - X)F_0$.

### Determining $IC_{50}$ values for various antibiotics

Single colonies were grown overnight as above and diluted 100-fold into M9 media. Antibiotic concentration gradients were created by serially diluting from 100 μg/ml, and included 0 μg/ml. Growth was measured in 96-well plates (Corning) with 200 μl liquid per well; wells were covered with 50 μl mineral oil (Sigma-Aldrich Chemicals) to prevent evaporation and density ($OD_{600}$) data were collected every 10 min using a Tecan M200 Infinite Pro plate reader at 37°C. $IC_{50}$ values were determined by first calculating growth rates by iteratively finding the linear region of increase; these growth rates were then fit to a Hill equation, where $\mu_{max}$, $n$, and $A$ are the maximum growth rates, Hill coefficient, and antibiotic concentration, respectively (equation 1):

$$\mu(A) = \frac{\mu_{max} IC_{50}^n}{IC_{50}^n + A^n} \tag{1}$$

### Luminescence reporter and experimental protocol

Overnight cultures were diluted 100-fold into M9 media supplemented with selection antibiotics; 1-ml aliquots were apportioned in 1.5-ml Eppendorf tubes, and experimental antibiotic concentrations were added as appropriate. Cultures were incubated at 30°C with shaking at 250 rpm for 2 h; cells were then resuspended in fresh M9 media containing only selection antibiotics; 200 μl aliquots were transferred to black 96-well plates, and time course luminescence measurements were taken using a Tecan Infinite M200 Pro plate reader every 10 min. Three replicates were used per experimental condition, and only alternate wells were used on 96-well plates to avoid capturing luminescence signal from neighboring wells. Time series data were normalized to initial conditions and population doubling time was determined relative to the initial time point.

### Fluorometric assay of efflux activity and modulation with CCCP

Efflux rates were measured by adapting an ethidium bromide (EtBr)-based assay for efflux pump activity (Viveiros *et al*, 2008; Paixão *et al*, 2009). At sub-toxic concentrations, ethidium bromide (EtBr) accumulates intracellularly, where it can be detected using plate reader fluorescence measurements (excitation/emission 530 nm/585 nm); previous studies have shown that this signal is significantly stronger intracellularly, as ethidium bromide binds to bacterial DNA. EtBr stock solution was prepared at 1 mg/ml in water and stored at room temperature. To measure efflux in BW25113, overnight cultures were washed and diluted 50-fold into a minimal M9 buffer media (M9 as described above, without glucose or casamino acid); EtBr was added to a final concentration of either 2 μg/ml. Efflux pump activity was inhibited by adding increasing concentrations of carbonyl cyanide *m*-chlorophenyl hydrazone (CCCP); stock solutions were prepared by dissolving

CCCP in dimethyl sulfoxide (DMSO) to a final concentration of 1 mg/ml and stored at 4°C. Cells were plated and covered with 50 μl mineral oil to prevent evaporation. Intracellular EtBr signal and cell density ($OD_{600}$) were measured every 10 min at 30°C using a Tecan InfintePro M200 plate reader (EtBr excitation/emission 530/585 nm). Fluorescence time courses were normalized to $OD_{600}$. Efflux rates were calculated assuming that intracellular EtBr concentration is dictated by the following equation:

$$\frac{dE_{in}}{dt} = k_{in}E_{out} - k_{out}E_{in},$$ (2)

where $E_{in}$ and $E_{out}$ represent the intracellular and extracellular EtBr concentrations, respectively. We assume $E_{total} = E_{in} + E_{out}$ to be constant, in accordance with Paixão et al (2009). Thus, equation (2) can be solved to yield:

$$E_{in}(t) = \frac{k_{in}}{k_{in}+k_{out}}E_{total} + \left(E_{in}(0) - \frac{k_{in}}{k_{in}+k_{out}}E_{total}\right)e^{-(k_{in}+k_{out})t}.$$ (3)

Time series fluorescence data were fit to equation (3), and efflux rates were determined, assuming $k_{out} = 0$ at high CCCP concentrations.

### Model development and assumptions

We modeled aminoglycoside action similarly to a previously published framework (Tan et al, 2012), using six ordinary differential equations (ODEs):

$$\frac{d[C]}{dt} = \frac{k_1[C]}{V_1+[C]} - k_f[C][A_{in}] - k_u[C] + k_b[CA],$$ (4)

$$\frac{d[A_{in}]}{dt} = k_{in}[A_{out}] - k_{out}[A_{in}] - k_f[C][A_{in}] + k_b[CA] + k_r[CA'],$$ (5)

$$[A_{out}] = A_0 \text{ if } t < D, \text{ otherwise } [A_{out}] = 0,$$ (6)

$$\frac{d[CA]}{dt} = k_f[C][A_{in}] - k_b[CA] - k_d[CA],$$ (7)

$$\frac{d[CA']}{dt} = k_d[CA] - k_r[CA'].$$ (8)

$$\frac{dN}{dt} = \mu_0\left(\frac{[C]}{V+[C]}\right)N\left(1 - \frac{N}{N_m}\right)$$ (9)

In this formulation, $[C]$ represents the free ribosome concentration, which is synthesized following Michaelis–Menten dynamics. Antibiotic shuttles between the extracellular ($[A_{out}]$) and intracellular ($[A_{in}]$) environment according to the rates of influx and efflux, $k_{in}$ and $k_{out}$, respectively. Ribosomes reversibly bind to intracellular antibiotic (rates $k_f$ and $k_b$) yielding the complex $[CA]$. This complex undergoes a two-step degradation process, releasing antibiotic to the intracellular concentration (Kaplan & Apirion, 1975; Edmunds & Goldberg, 1986; Zundel et al, 2009). $k_u$ represents the basal rate of ribosome turnover. Finally, the maximum population growth rate $\mu_0$ is scaled by the ribosome concentration (Neidhardt, 1996). All rate constants used are listed in Appendix Table S2.

For transient dosing simulations, we assumed that the extracellular antibiotic concentration $A_0$ did not significantly decrease during the dosing interval D. That is, $[A_{out}] = A_0$ for $t < D$, otherwise $[A_{out}] = 0$.

Recovery time was quantified on both the individual and population levels, both relative to the end of the treatment period. In the former case, we define the recovery time $RT_{cell}$ as the time required to achieve $0.5 * \max\left(\frac{d[C]}{dt}\right)$; in the latter, the recovery time $RT_{pop}$ is defined as the time required for the cell density to double.

### Periodic dosing simulations

Periodic dosing simulations were specified by the following parameters: the antibiotic concentration A, the duration of one dose $T_1$, the time between doses $T_2$, and the total number of doses $N_{dose}$. For each treatment, recovery time was determined relative to the end of the final dose; we denote the recovery time following $N_{dose}$ as $RT_n$. Thus, large values of $RT_n$ correspond to successful treatments, whereas minimal value of $RT_n$ indicates treatment failure. Therefore, the results presented in Fig 2 correspond to $RT_1$; as shown in Appendix Fig S1, we find that $RT_1$ can predict the efficacy of multi-dose treatments; our results indicate a sharp transition in treatment efficacy at $\frac{T_2}{RT_1} \sim 1$. That is, any treatment where doses are spaced more frequently than the corresponding single dose recovery time ($RT_1$) will remain effective.

### Modeling arbitrary dose profiles

To demonstrate the generality of the dependence between recovery time and total antibiotic, implemented both triangular and exponentially decaying dose profiles. For triangular doses, given the initial antibiotic concentration $A_0$ and duration D, we assumed a symmetric profile, with a maximum value of $A_0$ at $t = D/2$ For exponentially decaying doses, we assumed a rate constant such that $A_{out}(D) = 0.01A_0$.

### Comparing antibiotic mechanisms of action

We adopted a minimal set of ODEs capable of representing various mechanisms of action (equations 10–12, Appendix Table S2 shows parameter values used). Here, $[T]$ and $[A]$ represent the drug target and intracellular antibiotic concentration, respectively; $[P]$ represents the antibiotic–target binding product. $k_0$ represents the basal target synthesis rate, $k_f$ and $k_b$ represent the binding and dissociation rates between antibiotic and target, $k_p$ and $K_T$ represent the maximum synthesis and half-maximal values of $[T]$, $d_T$ and $d_p$ represent the degradation rates of target and antibiotic–target complex, respectively. Finally, $A_{out}$ represents the extracellular antibiotic concentration during a dose, which we assume to be constant; $k_{in}$ and $k_{out}$ represent the influx and efflux rates across the cell membrane, respectively. By setting various parameters to zero, these equations can be used to compare diverse modes of action. Setting $k_0 = 0$ results in target synthesis by positive feedback with rapid antibiotic-mediated degradation; setting $d_p = 0$ corresponds to antibiotic-inducing minimal target degradation; and setting $d_p = 0 = k_t$ corresponds to antibiotics whose targets are not subject to positive feedback. As with the full model, intracellular recovery time was calculated as the time to achieve $0.5 * \max\left(\frac{d[T]}{dt}\right)$ during recovery.

$$\frac{d[T]}{dt} = k_0 + \frac{k_t[T]^n}{K_t + [T]^n} - k_f[T][A] + k_b[P] - d_T[T] \tag{10}$$

$$\frac{d[A]}{dt} = k_{in}A_{out} - k_{out}[A] - k_f[T][A] + k_b[P] + d_p[P] \tag{11}$$

$$\frac{d[P]}{dt} = k_f[T][A] - k_b[P] - d_P[P] \tag{12}$$

## Data availability

MATLAB codes for all main figure simulation results are provided as Code EV1. The FAIRDOMHub investigation can be accessed at https://fairdomhub.org/investigations/155 and an SED-ML simulation reproducing Fig 2C, panel ii can be found at: https://jjj.bio.vu.nl/models/experiments/srimani2017_fig2cii/simulate.

**Expanded View** for this article is available online.

## Acknowledgements

The authors wish to acknowledge assistance from the Duke University Shared Materials Instrumentation Facility (SMIF) in fabricating the microfluidic platform, the Duke University Light Microscopy Core Facility (LMCF, particularly Dr. Sam Johnson) in carrying out experiments, Hannah Meredith for helpful discussions on PAE, and Dr. Kui Zhu for input and advice on efflux inhibition. This study was partially supported by the US Army Research Office (W911NF-14-1-0490, LY), the National Institutes of Health (2R01-GM098642, LY), a David and Lucile Packard Fellowship (LY), a Duke Center for Biotechnology and Tissue Engineering Fellowship (JKS), and the Howard G. Clark fellowship (AJL).

## Author contributions

JKS conceived the work, designed and carried out experimental and modeling studies and data analysis, and wrote the manuscript. SH designed and fabricated the microfluidic platform, assisted in experimental studies, and contributed to manuscript revisions. AJL fabricated microfluidic chips, assisted in carrying out experiments and data analysis, and contributed to manuscript revisions. LY conceived the work and assisted in research design, data interpretation, and writing and revising the manuscript.

## Conflict of interest

The authors declare that they have no conflict of interest.

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
