## [Review Process File · Molecular Systems Biology]

Drug detoxification dynamics explain the postantibiotic effect

Jaydeep K. Srimani, Shuqiang Huang, Allison J. Lopatkin & Lingchong You

Corresponding author: Lingchong You, Duke University

Review timeline:

Submission date:	30 April 2017
Editorial Decision:	07 June 2017
Revision received:	28 July 2017
Editorial Decision:	06 September 2017
Revision received:	19 September 2017
Accepted:	22 September 2017

Editor: Maria Polychronidou

Transaction Report:

1st Editorial Decision

07 June 2017

Thank you again for submitting your work to Molecular Systems Biology. We have now heard back from the two referees who agreed to evaluate your study. As you will see below, the reviewers raise a series of concerns, which we would ask you to address in a revision of the manuscript.

The reviewers' recommendations are rather clear so I think that there is no need to repeat the points listed below. Of course, feel free to contact me in case you would like to discuss any particular point in further detail.

REVIEWER REPORTS

Reviewer #1:

The paper of Srimani et al is an interesting and well written study of the post-antibiotic effect or PAE. It has been known for some time that particular antibiotics suppress bacteria growth even after a brief exposure. Historically, it has been observed that the extent of the delay depend on several proposed factors including the antibiotic, its mechanism of inactivation, the dose, the period of the exposure among others. Srimani et al look at each of these parameters and identify their relationship to the PAE. A model *E. coli* system is used with specific antibiotic streptomycin to parameterize and then probe the relationship of parameters like dose to the measured PAE. The model builds on earlier studies wherein ribosomes are the proxy that allows re-growth. That the rate of ribose degradation, altering the efflux rate, or increasing ribosome synthesis could all be linked in a

quantitative manner to the recovery time as measured in microfluidic devices is essential and allowed the authors to propose that efflux mediated recovery may be the general reason for the experimentally observed PAEs. This information could be used to better inform potential drug therapies and to know when to expect important PAE effects and when not to. For example as expected beta-lactams are not dependent on effluxers and thus showed little dependence. This is a good contribution to our ability to understand and model important effects of antibiotic and I recommend it for publication with some modest revisions.

Minor questions/suggestions:

1. On pp.6-7 the authors suggest that the effect they observe is "independent of initial cell density (Figure S2)." I wonder to what extent this is unique to the class of antibiotics being investigated. Antibiotics such as cationic antimicrobial peptides or cyclic lipopeptides that damage cell membranes might have a strong dependence on population since a small population has a smaller surface area whereas a large population has a larger surface area that could soak up more peptide and effectively reduce its killing power.
2. The authors should make clear how the mathematical model used in this work is significantly different compared to reference 23 (zur Wiesch et al, 2015). Zur Wiesch et al also use a similar model involving ribosomal replication but with kinetic parameters derived from the literature.
3. The authors suggest that their model is generally applicable to model many antibiotic classes. Authors classify antibiotics based on rate of ribosomal degradation/synthesis which is a key factor in the model and clearly true for the studied class of antibiotics. How is the model generalizable to drugs like Cipro etc?
4. On pp. 10-11 the authors examined three motifs of action and found that the efflux inhibitor was correlated strongly with PAE effects for three aminoglycosides but not for TET which I would think would be responsive as well. What is the explanation for Motif 2 not being well correlated? (Figure 4c)
5. What is the effect of a pulse train or spikes of doses as a function of frequency look like? The multi-dosing studies in-silico look very interesting with the phase transition behavior dependent on T2/RT1. The authors should consider testing the predictions using experiments.
6. While MIC is a terrible proxy for any quantitative modeling it would be useful to include the MIC or IC50 early on p 4 rather than in the Supplement S7. It helps to know where the dose is relative to this common metric.

It might be wise to insert at least one or two sentences into the Discussion recognizing the in vivo complexities of drug pharmacodynamics as usually discussed in the pk/pd literature. Presumably the general conclusion of antibiotic class and PAE are generally true but the actual utilization of the best combinations of doses and timing are also dependent on considerations of pk/pd.

Minor comments

- Equation S10 does not have a Hill coefficient for the feedback term for [T], it seems to be required based on supplementary fig S6, part C of the legend.
- Fig 3: Use of terminology: Total antibiotic (ug/ml * min) is confusing. Does it mean the same quantity (dimension) for both experimental data and model data? Maybe total antibiotic can be multiplied with flow rate (ml/min) to give total antibiotic mass in ug and report that as well?
- Figure 1A legend regarding the shading of antibiotic treatment in the graph is not correct.
- Figure 2B legend mentions panel 3 However, panel 3 is not labeled in the figure.
- Fig S1A: does not mention units of concentration of Streptomycin (ug/ml : mentioned in fig 1C)
- Fig S7 : it is not clear if x axis is in log scale.

Reviewer #2:

The manuscript presents mathematical modeling and substantial amount of experiments that explain the postantibiotic effect (PAE); temporary growth suppression after transient antibiotic treatment. It proposes a minimal unifying motif to account for PAE in response to various antibiotics and uses it to predict the effects of efflux inhibition-one of the key processes. While the simulation results still have some gaps with experimental observations, I find this model very useful to understand the PAE and explore different optimal strategies for existing drugs. I think this manuscript will appeal to the readers of MSB, but the text needs to be polished and more importantly, the authors should address

the following points:

1. In Figure 1A, the authors define the recovery time as double time of a population (cell density). However, authors use fluorescent protein expression to determine the recovery time from Figure 1C without any justification. Is this the same reporter mentioned at page 11?
2. I'm wondering if authors have thought about the potential caveat to using a translational reporter to estimate population density for different types of antibiotics including ribosomal targeting antibiotics. Can a cell express lower level of GFP at the removal of ribosomal targeting antibiotics ($t=0$, reference point) compared to later time points? Can this artificially shorten the recovery time at higher concentration of antibiotic treatment (by artificially lowering the reference cell density)?
3. I have some confusion in Figure 2D. The first sentence of Figure 2D figure legend is "Antibiotic efflux timescale sets intracellular recovery RT_{cell} ." However, as the authors mentioned in the main text, the detoxification time scale is set by the balance between three processes: efflux, binding, and degradation. If I understand correctly, Figure 2D is about detoxification, not about efflux. Am I missing something? Is there any reason to think as 'detoxification = antibiotic efflux'?
4. RT_{pop} and RT_{cell} correlation plot in Figure 2E inset should be either separate point (panel) or in supplementary. Since they claim single cell dynamics associated with population dynamics based on this plot both in abstract and discussion, I think authors should make this point clear rather than put it as a small inset.
5. It seems too bold to claim the "we show that population recovery time is a direct consequence of dynamics in individual cells" without any single cell data. Unless I missed something, the authors should refrain from such unnecessary and exaggerated statements.
6. What is the reasoning behind the use of specific antibiotics to validate key parameters in Figure 3? According to Figure 4, it seems very obvious that different mode of action can create range of PAE behavior. I guess there's a good reasoning behind using different antibiotics in Figure 3A and B; for instance, use an antibiotic without a degradation term to specifically test the effect of efflux inhibition? But there is no explanation anywhere. Also, since it is evident in Figure 4 that different modes of action can result in different PAE behavior, the authors should provide information about which antibiotics were used for Figure 3C simulation/experiments.
7. Why are Figure 3B and Figure 4A middle different?
8. What are the potential reasons of the discrepancy between Figure 4A and B especially for the middle case? Simulation results seem to be much closer to the left case, while experimental results look much similar to the right case. Can this be a hint to improve the model?

Minor points:

1. The left panel of Figure 1A is confusing. What is 'A' referring in blue shading area? Also, the relationship between '2' in y-axis and 'recovery time' should be clearer.
2. Figure legends are different from the figures in many places. For instance, there is no 'red shading' in Figure 1A.
3. Similarly, the main text misses Figure 4B and has comments about Figure 4C, which does not exist.
4. Supplementary Figure S5C figure legend is missing.
5. Supplementary Figure S6 uses colored lines without any explanation for each color.
6. It seems like CCCP never showed a negative effect. Can there be any reason to not use efflux pump inhibitors (EPIs) together?
7. Could this model be used to predict the efficacy of other antibiotic adjuvants?

1st Revision - authors' response

28 July 2017

We thank you and the reviewers for your time and consideration in evaluating our manuscript entitled "**Drug detoxification dynamics explain the postantibiotic effect**". The reviewers'

comments raised several points regarding the technical clarity and applicability of our conclusions to various antibiotics.

Here, we wish to emphasize the generality of our conclusion: antibiotic binding, efflux, and degradation dynamics are applicable to all drugs of various mechanisms of action, and as demonstrated by our work, are sufficient to explain the relationship between dosing parameters and subsequent population recovery. Our analysis provides an intuitive interpretation of the varying degrees of PAE in a variety of antibiotics and bacterial species described in past studies. Moreover, although the efficacy of efflux pump inhibitors (EPIs) may be modulated drug-specific rate parameters, we show that this strategy may be preferable under certain general regimes. These conclusions may be applicable to optimizing the selection of drug-adjuvant pairs for infectious disease treatment.

In light of these constructive comments and suggestions, we have carried out additional modeling and experimental analysis and revised our manuscript accordingly. These analyses include the correlation between fluorescence/luminescence reporters and true cell density, and the implications of periodic dosing regimen frequencies. Our major revisions are shown in blue in the manuscript.

We hope you and reviewers will find that our manuscript has been substantially improved and suitable for publication at *Molecular Systems Biology*.

Point-by-point responses to reviewers' comments

Reviewer #1:

The paper of Srimani et al is an interesting and well written study of the post-antibiotic effect or PAE. It has been known for some time that particular antibiotics suppress bacteria growth even after a brief exposure. Historically, it has been observed that the extent of the delay depend on several proposed factors including the antibiotic, its mechanism of inactivation, the dose, the period of the exposure among others. Srimani et al look at each of these parameters and identify their relationship to the PAE. A model *E. coli* system is used with specific antibiotic streptomycin to parameterize and then probe the relationship of parameters like dose to the measured PAE. The model builds on earlier studies wherein ribosomes are the proxy that allows re-growth. That the rate of ribose degradation, altering the efflux rate, or increasing ribosome synthesis could all be linked in a quantitative manner to the recovery time as measured in microfluidic devices is essential and allowed the authors to propose that efflux mediated recovery may be the general reason for the experimentally observed PAEs. This information could be used to better inform potential drug therapies and to know when to expect important PAE effects and when not to. For example as expected beta-lactams are not dependent on effluxers and thus showed little dependence. This is a good contribution to our ability to understand and model important effects of antibiotic and I recommend it for publication with some modest revisions.

We thank the reviewer for his/her time and detailed evaluation of our manuscript. The reviewer has accurately described our work, and has raised several constructive points/suggestions. We have addressed each of these points individually as follows, as well as in the revised manuscript.

Minor questions/suggestions:

1. On pp.6-7 the authors suggest that the effect they observe is "independent of initial cell density (Figure S2)." I wonder to what extent this is unique to the class of antibiotics being investigated. Antibiotics such as cationic antimicrobial peptides or cyclic lipopeptides that damage cell membranes might have a strong dependence on population since a small population has a smaller surface area whereas a large population has a larger surface area that could soak up more peptide and effectively reduce its killing power.

Indeed, a larger population would have a larger total surface area, and antibiotics that could be "soaked up" would potentially exhibit a dependence on initial density. However, cationic antimicrobial peptides or cyclic lipopeptides are significantly different from the antibiotics we used in our study. Here, we focus on small molecule drugs that are transported across the cell membrane

and bind to targets either in the cytoplasm (e.g. ribosome) or periplasm. The reviewer rightly points out that inoculum effect is an intrinsic property for some of these antibiotics. In our experiments, we have used sufficiently low initial cell densities such that populations were effectively inhibited over the range of dose conditions tested.

We have revised the text to clarify this point (**page 8**).

2. The authors should make clear how the mathematical model used in this work is significantly different compared to reference 23 (zur Wiesch et al, 2015). Zur Wiesch et al also use a similar model involving ribosomal replication but with kinetic parameters derived from the literature.

Indeed, our model is similar to that of zur Wiesch et al (zur Wiesch et al, 2015) but there are a number of differences. We adapted our full model (**Figures 2-3**) from (Tan et al, 2012). This full model precedes the Zur Wiesch model and provides a more detailed description of antibiotic dynamics. Our full model has been validated by experimental data generated both from our previous study and the current one, as well as relevant literature data. Moreover, our model does not rely on effective antibiotic dilution due to bacterial growth to achieve subinhibitory intracellular concentrations. Instead, our model emphasizes the effects of intracellular binding and degradation dynamics, as well as the efflux dynamics of the antibiotics. Moreover, unlike the zur Wiesch model, we do not consider the contribution from non-specific antibiotic binding. Though not critical for PAE generation, the non-specific binding would serve to extend PAE. Our simplified models used in **Figure 4** are designed to compare how three different antibiotic motifs can generate PAE, and how this relationship can be modulated by EPIs. These models further emphasize the importance of intracellular processes, e.g. target titration and antibiotic export.

We have revised the text to clarify the differences between our model and previous studies (**page 15**).

3. The authors suggest that their model is generally applicable to model many antibiotic classes. Authors classify antibiotics based on rate of ribosomal degradation/synthesis which is a key factor in the model and clearly true for the studied class of antibiotics. How is the model generalizable to drugs like Cipro etc?

We thank the reviewer for this insightful question. Our model is generalizable to various antibiotics because the key processes (influx/efflux, target binding, and degradation) are generally applicable to different antibiotics. In Figure 4, we demonstrate that various intracellular antibiotic-target motifs can indeed lead to recovery times that depend on total antibiotic. Here, we do not assume that targets necessarily correspond to ribosomes. Indeed, fluoroquinolones, e.g. ciprofloxacin, which act as DNA synthesis inhibitors, would fall under the second motif – they inhibit a positive feedback loop without inducing rapid target degradation. Similarly, β -lactams fall under the third motif. Moreover, we note that our experimental results (**Figure 4B**) confirm the effects of efflux inhibitors on every antibiotic tested.

We have revised the text to clarify and emphasize this point (**page 12**).

4. On pp. 10-11 the authors examined three motifs of action and found that the efflux inhibitor was correlated strongly with PAE effects for three aminoglycosides but not for TET which I would think would be responsive as well. What is the explanation for Motif 2 not being well correlated? (Figure 4c).

We thank the reviewer for noting this difference. As demonstrated in our previous study (Tan et al, MSB 2012), tetracycline does not induce rapid ribosome degradation. For such an antibiotic (**Figure 4A, middle panel**), our model predicts that the addition of EPI would not significantly extend the recovery time. This property is consistent with other antibiotics (e.g. chloramphenicol) that also do not induce rapid target degradation.

5. What is the effect of a pulse or spikes of doses as a function of frequency look like? The multi-dosing studies in-silico look very interesting with the phase transition behavior dependent on T2/RT1. The authors should consider testing the predictions using experiments.

We thank the reviewer for appreciating the applicability of our conclusions to periodic dosing optimization. In light of this comment, we have carried out additional analysis (**Figure S5D**). The new results suggest that dosing outcome could be controlled by rationally choosing antibiotic concentration and frequency based on *a priori* knowledge of population response to a single dose. Furthermore, **Figure S5D** shows that populations exhibit a biphasic recovery time in response to increasing frequency. That is, populations survive better (lower recovery times) at intermediate dosing frequencies. This is consistent with previously published work (Tan et al, 2012). We agree that experimental tests of some of these model predictions would be valuable. However, we feel that such tests are better suited for future work as the current work is focused on establishing the kinetic basis of the PAE generation *per se*.

6. While MIC is a terrible proxy for any quantitative modeling it would be useful to include the MIC or IC₅₀ early on p 4 rather than in the Supplement S7. It helps to know where the dose is relative to this common metric.

We thank the reviewer for this helpful suggestion. We have revised the main text to include the IC₅₀ values in the main text for both streptomycin (**page 6**) and chloramphenicol (**page 10**).

7. It might be wise to insert at least one or two sentences into the Discussion recognizing the *in vivo* complexities of drug pharmacodynamics as usually discussed in the pk/pd literature. Presumably the general conclusion of antibiotic class and PAE are generally true but the actual utilization of the best combinations of doses and timing are also dependent on considerations of pk/pd.

We thank the reviewer for this valuable advice. We have updated the Discussion section to reflect the PK/PD and bioavailability factors that commonly affect *in vivo* antibiotic dosing protocols, and how our analysis can be applied to the broader subject of rational dosing design (**pages 14-15**).

Minor comments

- Equation S10 does not have a Hill coefficient for the feedback term for [T], it seems to be required based on supplementary fig S6, part C of the legend.

We thank the reviewer for pointing out this inconsistency and have corrected this omission in the revised text.

- Fig 3: Use of terminology: Total antibiotic (ug/ml * min) is confusing. Does it mean the same quantity (dimension) for both experimental data and model data? Maybe total antibiotic can be multiplied with flow rate (ml/min) to give total antibiotic mass in ug and report that as well?

Here, total antibiotic refers to total exposure for a given dose, i.e. concentration and duration pair. It is the area under the curve (AUC) for the dose, which for rectangular doses is simply concentration multiplied by duration, hence the units of $\mu\text{m}/\text{mL} \cdot \text{min}$. As the reviewer points out, this is indeed the same quantity for both modeling and experimental results; we have clarified the text to reflect this point. Moreover, we have refrained from using the flow rate to give the total antibiotic in μg , as this suggests that that amount was accumulated in the environment, whereas continuous flow was utilized for all microfluidic experiments. We have modified the labeling to use the phrase “Total antibiotic exposure” throughout the text, to clarify this point.

- Figure 1A legend regarding the shading of antibiotic treatment in the graph is not correct.

We thank the reviewer for pointing out this inconsistency and have revised the figure legend to reflect the shading in Figure 1A.

- Figure 2B legend mentions panel 3 However, panel 3 is not labeled in the figure.

We thank the reviewer for pointing out this inconsistency and have revised the figure legend to refer to 2 panels, as in the figure.

- Fig S1A: does not mention units of concentration of Streptomycin (ug/ml : mentioned in fig 1C).

We thank the reviewer for pointing out this omission and have revised the figure legend to specify the units of streptomycin concentration used.

- Fig S7 : it is not clear if x axis is in log scale.

We have revised the figure legend to specify that the x-axis shows antibiotic concentrations on a log scale.

Reviewer #2:

The manuscript presents mathematical modeling and substantial amount of experiments that explain the postantibiotic effect (PAE); temporary growth suppression after transient antibiotic treatment. It proposes a minimal unifying motif to account for PAE in response to various antibiotics and uses it to predict the effects of efflux inhibition—one of the key processes. While the simulation results still have some gaps with experimental observations, I find this model very useful to understand the PAE and explore different optimal strategies for existing drugs. I think this manuscript will appeal to the readers of MSB, but the text needs to be polished and more importantly, the authors should address the following points:

We thank the reviewer for his/her detailed consideration of our work, and the appreciation of its applicability towards optimized dosing. The reviewer brings up a number of important questions/suggestions that will improve the quality and rigor of our conclusions; we have addressed these points individually as follows, noting the relevant changes in the revised text and figures.

1. In Figure 1A, the authors define the recovery time as double time of a population (cell density). However, authors use fluorescent protein expression to determine the recovery time from Figure 1C without any justification. Is this the same reporter mentioned at page 11?

In Figures 1-3, we use constitutive GFP expression as the surrogate measure of the cell density. The use of GFP as a readout is better suited to microscopy, as it allows for high-resolution quantification based on imaging. Accounting for GFP signal from non-growing cells does not affect the log-linear feature of the observed antibiotic dependence (**Figure S2**). In Figure 4, we use a constitutively expressed bioluminescence signal as a reporter of cell viability. As explained in the text, bioluminescence has often been used to measure living cells, and bypasses the need to control for cell death as in the microfluidic device (**Figure S2**). Moreover, luminescence is better suited for plate-reader measurements, thanks to the minimal background luminescence generated by the culture medium. We have included additional data (**Figure S1**) to calibrate both GFP and luminescence reporters with CFU counts. These data demonstrate that both readouts increase approximately linearly with cell density.

We have revised the text to emphasize this readout relationship (**pages 6, 13**).

2. I'm wondering if authors have thought about the potential caveat to using a translational reporter to estimate population density for different types of antibiotics including ribosomal targeting antibiotics. Can a cell express lower level of GFP at the removal of ribosomal targeting antibiotics ($t=0$, reference point) compared to later time points? Can this artificially shorten the recovery time at higher concentration of antibiotic treatment (by artificially lowering the reference cell density)?

Indeed, it is possible for antibiotic treatment to alter the GFP expression in individual cells; this would be most evident under strong antibiotic treatment, i.e., higher concentrations and longer durations. To address this possibility, we have included additional data (**Figure S1**) to show the

correlation between GFP expression and CFU counts. These results indicate that fluorescence signal remains a good surrogate of cell density in the presence of antibiotic treatment.

We have revised the text to reflect this point (**page 6**).

3. I have some confusion in Figure 2D. The first sentence of Figure 2D figure legend is "Antibiotic efflux timescale sets intracellular recovery RT_{cell} ." However, as the authors mentioned in the main text, the detoxification time scale is set by the balance between three processes: efflux, binding, and degradation. If I understand correctly, Figure 2D is about detoxification, not about efflux. Am I missing something? Is there any reason to think as 'detoxification = antibiotic efflux'?

We thank the reviewer for raising this distinction. The reviewer is correct that a combination of efflux, binding, and degradation contributes to setting the recovery time of a population, which is what we intend to convey. We have revised the main text and figure legend to be more precise in our language (**page 8**).

4. RT_{pop} and RT_{cell} correlation plot in Figure 2E inset should be either separate point (panel) or in supplementary. Since they claim single cell dynamics associated with population dynamics based on this plot both in abstract and discussion, I think authors should make this point clear rather than put it as a small inset.

We thank the reviewer for this suggestion. We have revised **Figure 2** to include the correlation between RT_{pop} and RT_{cell} as a main panel. We also note that **Figure S4** further demonstrates the strong correlation between these two metrics over a range of ribosome degradation rates.

5. It seems too bold to claim the "we show that population recovery time is a direct consequence of dynamics in individual cells" without any single cell data. Unless I missed something, the authors should refrain from such unnecessary and exaggerated statements.

We thank the reviewer for raising this concern. As the reviewer notes, our study does not include any single cell experimental measurements of recovery time. We have removed these statements from the revised text.

6. What is the reasoning behind the use of specific antibiotics to validate key parameters in Figure 3? According to Figure 4, it seems very obvious that different mode of action can create range of PAE behavior. I guess there's a good reasoning behind using different antibiotics in Figure 3A and B; for instance, use an antibiotic without a degradation term to specifically test the effect of efflux inhibition? But there is no explanation anywhere. Also, since it is evident in Figure 4 that different modes of action can result in different PAE behavior, the authors should provide information about which antibiotics were used for Figure 3C simulation/experiments.

We thank the reviewer for bringing up this need for clarification. Indeed, as mentioned by the reviewer, in **Figure 3A/D** we compare streptomycin and chloramphenicol treatment to demonstrate the impact of fast and slow ribosome turnover/degradation, respectively. As pointed out in the main text, previous work in our lab has shown that streptomycin (and not chloramphenicol) induces a heat shock response (HSR), resulting in increase protease expression and subsequent ribosome turnover. This is independent of any efflux rate modulation.

In **Figure 3B/E** we examine the effect of modulating efflux rate (using the efflux pump inhibitor CCCP *in vitro*). In this case, we compared recovery times in the presence and absence of CCCP in response to chloramphenicol treatment. Intuitively, we expected that the addition of an EPI would increase recovery time; therefore, we chose to use an antibiotic that alone induced a minimal recovery time. Finally, in **Figure 3C/F** we modulated the ribosome synthesis rate by varying the media conditions; for these experiments, we used streptomycin treatment. The logic here was that faster ribosome synthesis would lower recovery times, hence we chose an antibiotic that alone induced a significant dose-dependent recovery.

We have revised the text and figure captions to clarify and emphasize the logic of these choices (pages 9-11).

7. Why are Figure 3B and Figure 4A middle different?

We thank the reviewer for bringing up this point. The difference results from the use of two different models (also see our Response #2 to Reviewer 1). **Figure 3B** is generated from our full model, which provides a more detailed description of the reactions involved in bacterial responses to antibiotics targeting ribosomes. **Figure 4A** is generated from a simplified model; it is intended to examine the generation and modulation of PAE in a more general setting. We note that the qualitative trends from **Figure 3B and Figure 4A (middle)** are consistent: in both cases, our results indicate that in the case of slow target degradation, efflux inhibition does not result in significant increases in recovery time.

8. What are the potential reasons of the discrepancy between Figure 4A and B especially for the middle case? Simulation results seem to be much closer to the left case, while experimental results look much similar to the right case. Can this be a hint to improve the model?

We thank the reviewer for raising this important point. Indeed, the experimental results suggest that target degradation rates can lead to qualitative different outcomes: when degradation is rapid, efflux inhibition significantly increases recovery time (**Figure 4B left column**), whereas this effect is less distinct when degradation is slow (Figure 4B right column). In light of the reviewer's suggestion, we have optimized the models in terms of choice of parameter values, and updated the simulation results accordingly.

Minor points:

- The left panel of Figure 1A is confusing. What is 'A' referring in blue shading area? Also, the relationship between '2' in y-axis and 'recovery time' should be clearer.

In **Figure 1A**, 'A' refers to the duration of antibiotic treatment. We have revised the figure legend to clarify this point. In addition, we have modified the figure to emphasize the definition of recovery time as the time required for a two-fold increase in density.

- Figure legends are different from the figures in many places. For instance, there is no 'red shading' in Figure 1A.

We thank the reviewer for pointing out this inconsistency and have revised the relevant figure legends and text references for clarify these differences.

- Similarly, the main text misses Figure 4B and has comments about Figure 4C, which does not exist.

We have revised the main text to refer to **Figure 4B** only.

- Supplementary Figure S5C figure legend is missing.

We thank the reviewer for pointing out this omission and have added the legend for **Figure 5C**.

- Supplementary Figure S6 uses colored lines without any explanation for each color.

We have revised the figure (now **Figure S7**) using a color gradient to indicate increasing parameter values. The arrows in both panels also indicate the direction of parameter increase, as we have clarified the figure legend to reflect this point.

- It seems like CCCP never showed a negative effect. Can there be any reason to not use efflux pump inhibitors (EPIs) together?

If the reviewer is referring to the growth impact of CCCP, then indeed, we chose the CCCP concentration to subinhibitory (below the IC_{50}). In that case, changes in recovery time are attributable to efflux modulation as opposed to growth inhibition, or a combination thereof (**Figure 3E** and **Figure S6**).

If the reviewer is referring to the effect of CCCP on recovery time, the extent of this effect depends on the particular antibiotic and dose used (as illustrated in **Figure 4B**). The reviewer is correct that multiples EPIs could potentially be used, assuming that concentrations were chosen so as to not avoid synergistic growth inhibition. We have revised the main text to address this possibility. In our study, we chose CCCP, as it is one of the best-characterized EPIs, and has been shown to inhibit efflux of many antibiotics in a variety of bacterial species.

- Could this model be used to predict the efficacy of other antibiotic adjuvants?

Indeed, our model could be used to predict the effect of a variety of adjuvants, provided that they could be mapped to changes in one or more system parameters. For example, an adjuvant that increased ribosome degradation (and would therefore increase recovery time in response to a given antibiotic dose) could be modeled as increase in that particular rate parameter, as shown in Figure 3A. We sought to examine the effect of EPIs, as they have been the focus of many previous studies into increasing the efficacy of antibiotic treatments.

References

Tan C, Smith RP, Srimani JK, Riccione KA, Prasada S, Kuehn M, You L (2012) The inoculum effect and band-pass bacterial response to periodic antibiotic treatment. **8**: 1-11

zur Wiesch PA, Abel S, Gkatzis S (2015) Classic reaction kinetics can explain complex patterns of antibiotic action. *Science Translational Medicine* **7**: 287ra273-287ra273

2nd Editorial Decision

06 September 2017

Thank you again for submitting your work to Molecular Systems Biology. I have now had the chance to read the manuscript and your responses to the reviewers' comments. We are satisfied with the modifications made and we think that the manuscript is now suitable for publication.

Before we formally accept the manuscript for publication, we would ask you to address the following issues:

- We recently implemented a "model curation service" for papers that contain mathematical models. This is done together with Prof. Jacky Snoep and the FAIRDOM team and it is still in a pilot phase (therefore has not yet been announced officially). In brief, the aim is to enhance reproducibility and add value to papers containing mathematical models. Jacky Snoep's summary (*Model Curation Report*) is pasted below. As you most likely know already from your email exchange with him, and will also see in the report below, there are some minor issues, which we would ask you to fix when you submit your revision.

- In the Authors Checklist you mention (sections 21 and 22), that datasets and computational models will be available upon request. We would ask you to make sure that you provide all relevant data and computational models as EV Datasets or EV Computer code. All information related to datasets/code should be listed in a Data Availability section after the Materials and Methods in the main text. Please update the checklist accordingly.

Model Curation Report

Technical curation for the mathematical models in MSB-17-7723R

The equations used in the model for the study are well described in the manuscript, but some information was missing to reproduce the simulations shown in the manuscript. I contacted the authors and they quickly responded and gave the missing information, and also the Matlab files that were used for the simulations. In addition the authors noted that they did not supply the newest simulation results in the revised manuscript and provided with an updated figure file.

The information that was missing was: 1) initial values for the variables in the model, these should be provided, for instance in the supplementary material of the manuscript, and 2) the ranges of concentrations of antibiotics that were tested, and the ranges of incubation times that were used for the antibiotics. This information is available in the Matlab file, but I would recommend including these ranges in the figure legends for the respective model simulations.

After the authors provided us with the missing information we were able to reproduce the model simulations shown in the manuscript. We coded both the 6-variable model and the 3-variable model described in the manuscript and will make the SBML files for these models available, if the manuscript is accepted. In addition we have made a SED-ML script for the simulation shown in Fig 2Cii, which reproduces the manuscript figure in a life simulation in a web browser. This simulation script will also be made available if the manuscript is accepted. We could reproduce the model simulations shown in Figures 3 and 4 using the Matlab files that were made available by the authors.

2nd Revision - authors' response

19 September 2017

We thank you and the reviewers for your time and consideration in evaluating our revised manuscript entitled “**Drug detoxification dynamics explain the postantibiotic effect**”. We also appreciate the comments and feedback from Dr. Snoep and the FAIRDOM team. In light of their suggestions, we have made the following changes:

- We have included the relevant antibiotic concentration and dose duration values used in generating modeling results in the Figure 2 and Figure 4 captions.
- We have revised the Appendix to include all initial conditions for each set of simulations.
- We have included all computer code as EV materials, and revised the main text accordingly.

3rd Editorial Decision

22 September 2017

Thank you again for sending us your revised manuscript. We are now satisfied with the modifications made and I am pleased to inform you that your paper has been accepted for publication.

Corresponding Author Name: LINGCHONG YOU

Journal Submitted to: MOLECULAR SYSTEMS BIOLOGY

Manuscript Number: MSB-17-7723